

# Bayesian deconstruction of climate sensitivity estimates using simple models: implicit priors, and the confusion of the inverse.

James Annan and Julia Hargreaves

BlueSkiesResearch.org.uk
Settle, UK

*Correspondence to:* James Annan (jdannan@blueskiesresearch.org.uk)

**Abstract.**

Observational constraints on the equilibrium climate sensitivity have been generated in a variety of ways, but the epistemic basis of these calculations have not always been clearly presented and a number of results have been calculated which appear to

be based on somewhat informal heuristics. This causes a lack of clarity about the status of such results and how they compare to other analyses, in particular whether the differences between them may be due to differences in unstated assumptions rather than observational evidence.

In this paper, we show how these problems can be resolved. We demonstrate that many of these estimates can be reinterpreted within the standard subjective Bayesian framework in which a prior over the uncertain parameters is updated through a

likelihood arising from observational evidence. In many of these cases, the prior which was (under this interpretation) implicitly used exhibits some unconventional and possibly undesirable properties. We present alternative calculations which use the same observational information to update a range of explicitly presented priors.

Our calculations suggest that the heuristic methods do often generate reasonable results, in that they agree fairly well with the explicitly Bayesian approach using a reasonable prior. However, we also find some significant differences and argue that

the explicitly Bayesian approach is preferred, as it both clarifies the rôle of the prior, and allows researchers to transparently test the sensitivity of their results to it.

## 1 Introduction

While numerous explicitly Bayesian analyses of the equilibrium climate sensitivity have been presented (e.g. Tol and De Vos, 1998; Olson et al., 2012; Aldrin et al., 2012), many results have also been generated which appear to be based on more heuristic

methods. In this paper we examine several such estimates and demonstrate how they can be reinterpreted in the context of the subjective Bayesian framework, revealing in each case an underlying prior which can be deemed to have been implicitly used. That is to say, we present an explicitly Bayesian analysis which takes the same observational data together with the same assumptions/model underlying the data-generating process, which (when used to update this implicit prior), precisely replicates the published result. In some cases these implicit priors exhibit rather unconventional properties, and we argue that they are





unlikely to have been chosen deliberately, and would probably not have been used if the authors had presented a transparently Bayesian analysis. We rerun some of these analyses in a standard Bayesian framework, using the same observational evidence to update a range of explicitly stated priors. While in many cases these results are broadly similar to the existing published results, some differences will be apparent.

The paper is organised as follows. In Section 2 we introduce some concepts in Bayesian analysis which underpin our presentation. In Section 3, we explore several calculations in which researchers have estimated the climate sensitivity via direct calculation based on observationally-derived probability density functions, considering paleoclimate research (Annan and Hargreaves, 2006; Köhler et al., 2010; Rohling et al., 2012), the observational record of warming over the 20th century warming (Gregory et al., 2002; Mauritsen and Pincus, 2017), and analyses of interannual variability (Forster and Gregory,

2006; Dessler and Forster, 2018) in turn. We present a Bayesian interpretation of these calculations, and give some alternate analyses based on alternative, explicitly stated, priors. We argue that this latter approach is preferred, as it both clarifies the rôle of the prior, and allows researchers to transparently test the sensitivity of their results to it. We conclude with a general discussion about our results.

## 2 Principles and methods

### 2.1 Confidence intervals, Bayesian probability and the "confusion of the inverse"

Let us assume we have a measuring process that produces an observational estimate $x_o$ of an unknown (but assumed constant) parameter which takes the value $x_T$, with an observational error $\epsilon$ that can be considered a random draw from a specified error distribution, typically taken to be Gaussian:

$$x_o = x_T + \epsilon \tag{1}$$

where $\epsilon \sim N(0, \sigma)$. For simplicity, we assume here $\sigma$ is known. This "measurement model" is fundamental to analysis of observations in many scientific domains. For example, in climate science, analyses of observed global temperature anomalies are commonly generated and presented in this form.

  Following on from this measurement model, there is a simple syllogism (i.e. a logical argument that seems common in many areas of scientific research, which runs as follows: since we know *a priori* that $p(-2\sigma < \epsilon < 2\sigma) \simeq 95\%$, we can also write

*a posteriori* that $p(x_o - 2\sigma < x_T < x_o + 2\sigma) \simeq 95\%$ once $x_o$ is known. For example, if $\sigma = 0.25$ is given, and we observe the value $x_o = 74.60$ then the researcher may assert "there is $\sim 95\%$ probability that $x_T$ lies in the interval $(74.10, 75.10)$" or simply present a full probability density: "the pdf of $x_T$ is $N(x_o, \sigma) = N(74.60, 0.25)$".

  This syllogism is intuitively appealing but incorrect. It appears to arise from the misinterpretation of frequentist confidence intervals, as being Bayesian credible intervals. We should note that calculating and presenting the interval $x_o \pm 2\sigma$ as a fre-

quentist 95% confidence interval is an entirely valid procedure. That is to say, if we were to repeatedly take a new observation according to Equation 1, with each sample having an independent observational error, and generate the corresponding interval $(x_o - 0.5, x_o + 0.5)$ then approximately 95% of the intervals so generated would include the true value $x_T$. However, frequentist





confidence intervals are not the same thing as Bayesian credible intervals. The latter interpretation for an interval refers to a degree of belief that the particular interval that has been generated on a specific occasion does in fact include the parameter. Climate scientists are far from unique in this misinterpretation, which appears to be widespread throughout the scientific community (Hoekstra et al., 2014). Because this misunderstanding is so deeply embedded in scientific practice and discourse, we

now discuss and explain it in some detail.

We start by noting that probabilistic statements concerning the true value $x_T$ demand the use of the Bayesian paradigm, wherein the language and mathematics of probability may be applied to events that are not intrinsically random, but about which our knowledge is uncertain (Bernardo and Smith, 1994). The parameter $x_T$ here does not itself have a probability distribution; it was assumed to take a fixed value. Therefore to even talk of "the pdf of $x_T$" in this manner is to commit a

category error. It is the researcher's beliefs concerning $x_T$ that are uncertain, and this uncertainty is represented as their pdf for $x_T$.

Bayes' Theorem is a simple consequence of the axioms of probability: the joint density $p(x_o, x_T)$ of two variables $x_o$ and $x_T$ can be decomposed in two different ways via

$$p(x_o, x_T) = p(x_T|x_o)p(x_o) = p(x_o|x_T)p(x_T)$$

and thus

$$p(x_T|x_o) = p(x_o|x_T)p(x_T)/p(x_o). \tag{2}$$

$p(x_T|x_o)$ is our posterior density for the true value $x_T$ given the observational evidence $x_o$. $p(x_o|x_T)$ is commonly termed the 'likelihood' and is determined by the measurement model: for example, in the case of an unbiased Gaussian observational error, such as in Equation 1, the functional form of $p(x_o|x_T)$ is given by

$$p(x_o|x_T) = \frac{1}{\sqrt{2\pi}\sigma}e^{\frac{-(x_o-x_T)^2}{2\sigma^2}}.$$

When the terms for $x_o$ and $\sigma$ are replaced in this function by their known numerical values, this function looks like it could be a probability distribution for $p(x_T|x_o)$, but as Bayes' Theorem (Equation 2) makes clear, it is not in general the posterior pdf, instead being merely one term in its calculation. This is the critical point which underpins the analyses presented in this paper: the sampling distribution of the observation defined by measurement models such as Equation 1 directly defines the likelihood

$p(x_o|x_T)$ and not the posterior pdf $p(x_T|x_o)$.

The error in the syllogism is to interpret $p(x_o|x_T)$ as $p(x_T|x_o)$: this is a common fallacy known as the "confusion of the inverse" which is closely related to the "prosecutor's fallacy", the latter term generally being used in discrete probability where the phenomenon is more widely known and well studied. The fallacy is perhaps easiest to illustrate with discrete cases which compare $P(A|B)$ to $P(B|A)$ for a pair of events $A$ and $B$. For example, the probability of a person suffering from a rare

disease (event $A$), given that they tested positive for it (event $B$), is in general different from (and often rather lower than than) the probability that someone produces a positive test result, given that they are suffering from the disease. It has been known for some time that medical doctors routinely commit this transposition error (Gigerenzer and Hoffrage, 1995). Additional examples and discussion in relation to interval estimation can be found in Morey et al. (2016).





We now present a simple example in which the syllogism leads to poor results in a physically-based scenario with continuous data. We take as given that the timing error of a hand-held stopwatch is $\pm 0.25$s at one standard deviation (Hetzler et al., 2008). That is to say, the measured time $t_o$ is related to the true time, $t_T$, via $t_o = t_T + \epsilon$ with $\epsilon \sim N(0.0.25)$ (cf Equation 1). Let us consider an experiment in which an adult male colleague holds a dense object (say, a stone) at head height while standing, and

drops it while the experimenter times how long it takes for the stone to reach the ground.

An observed time of $t_o = 0.60$s could lead someone to say via the confusion of the inverse fallacy that the true time taken is represented by the Gaussian pdf $t_T \sim N(0.6, 0.25)$ (albeit with an assumed truncation at zero which we ignore for convenience). One implication of this pdf is that there is a 16% chance that the true time is less than 0.35s, and also a 16% chance that it is more than 0.85s. Ignoring the negligible air resistance and using the simple equation of motion under gravity $h = \frac{1}{2}at^2$,

one would have no choice but to conclude from these values that the experimenter's colleague has a 16% chance of being less than 60cm tall, and also a 16% chance of being greater than 4.5m tall. For a typical adult male, neither of these cases seems reasonable. We have obtained a measurement which is entirely unremarkable, with the observed time corresponding to a fall of around 1.75m. And yet the commonplace interpretation of an imprecise measurement as directly giving rise to a probability distribution for the measurand has lead to palpably ridiculous results. While in many cases the results will not be so silly, this

simple example does demonstrate that the methodology cannot be sound. More pernicious cases are where the interpretation is not so obviously silly and thus may be confidently presented, even though the methodology is still (as we have just shown) invalid.

In order to make sensible use of this observation, we can instead perform a simple Bayesian updating procedure. The distribution $N(0.6, 0.25)$ is actually correctly interpreted as the likelihood of the observed time $p(t_o|t_T)$, which can be used

to update a prior estimate. The distribution of adult male heights in the UK (in metres) is taken to be by $N(1.75, 0.07)$ and we use this as our prior. The drop time $t$ predicted from a height drop $h$ is given by $t = \sqrt{2h/a}$ where $a = 9.8ms^{-2}$ is the acceleration due to gravity. Due to the substantial observational uncertainty, the likelihood of the drop time is virtually flat across the support of the prior, varying by less than 1% across the range of 1.60m to 1.90m. The posterior estimate obtained through Bayes' Theorem is easily calculated by direct numerical integration and still approximates to $N(1.75, 0.07)$ to two

decimal places. The correct interpretation of the experiment is not, therefore, that the measurement shows there is a substantial probability of the researcher breaking a height record, but rather that the measurement is so imprecise that it does not add any significant information on top of what was already known.

While it is formally invalid, we must acknowledge that this syllogism does actually work rather well in many cases. In particular, if the likelihood $p(x_o|x_T)$ is non-negligible over a sufficiently small neighbourhood of $x_o$ such that a prior can

reasonably be used which is close to uniform in this region of $x_o$, then the true posterior calculated by a Bayesian analysis will be close to that asserted by the syllogism. For example, if the Gaussian prior $x_T \sim N(100, 20)$ were to be used in the original example, then when this is updated by the likelihood corresponding to the observation $x_o = 74.6$ with uncertainty $\sigma = 0.25$, the correct posterior $p(x_T|x_o)$ is actually given by $N(74.6, 0.25)$ to several significant digits. In the limiting case where an unbounded uniform prior is used for $x_T$, the syllogism is precisely correct.





Thus in practice the syllogism can often be interpreted as Bayesian analysis in which a uniform prior has been implicitly used, and in cases where this is reasonable it will generate perfectly acceptable results. Statements to this effect have occasionally appeared in some papers where a non-Bayesian analysis has been presented as directly giving rise to a posterior pdf. It may therefore seem that the terminology of 'fallacy' and 'confusion' is somewhat melodramatic: this convenient shortcut is often

harmless enough. However this cannot be simply asserted without proof: there are many examples of procedures for generating frequentist confidence intervals where the results cannot plausibly be interpreted as Bayesian credible intervals (Morey et al., 2016). As well as concerns over the prior, it is also essential when taking this shortcut that the observational uncertainty $\sigma$ is taken to be a constant which does not vary with the parameter of interest $x_T$. This may be the case when we consider uncertainties arising solely from an observational instrument, but is less clear when $\sigma$ includes a contribution from the system

under study. For example, if the uncertainty in an observed estimate of the forced temperature response in an analysis of climate change includes a contribution due to the internal variability of the climate system, then this internal variability could be expected to vary with the parameters of the system. In this case, the confusion of the inverse cannot be rescued by invocation of a uniform prior. However we do not explore this uncertainty in $\sigma$ further in this paper.

Some have attempted to retrospectively defend the use of this syllogism with the claim that the uniform prior is necessarily

the correct one to use, generally via the belief that this represents some sort of pure or maximal state of ignorance. However, it is well-established (and indeed is sometimes used as a specific point of criticism) that there is no such thing as pure ignorance within the Bayesian framework. See Annan and Hargreaves (2011) for further discussion of this in the context of climate science. As Morey et al. (2016) states: "Using confidence intervals as if they were credible intervals is an attempt to smuggle Bayesian meaning into frequentist statistics, without proper consideration of a prior." There is also a strand of

Bayesianism which asserts more broadly that in any given experimental context there is a single preferred prior, typically one which maximises the influence of the likelihood in some well-defined manner. Jeffreys Prior is one common approach within this "objective Bayesian" framework. However, it has the disadvantage that it assigns zero probability to events that the observations are uninformative about. This 'see no evil' approach does have mathematical benefits but it is hard to accept as a robust method if the results of the analysis are intended to be of practical use. In the real world, our inability to (currently) observe

something cannot rationally be considered sufficient reason to rule it out. We do not consider "objective Bayesian" approaches further.

## 2.2 Priors for the climate sensitivity

Most probabilistic estimates of the equilibrium climate sensitivity which have explicitly presented a Bayesian framework, have used a prior which is uniform in sensitivity $S$. There does not appear to be any principled basis for this choice, which has

been argued on the basis that it represented 'ignorance'. One could just as easily (and erroneously) argue that a prior which is uniform in feedback $\lambda = F_{2\times}/S$ was ignorant (here $F_{2\times}$ is the forcing arising from a doubling of $CO_2$). In fact both of these improper priors can exhibit a pathology which causes problems with their use. In particular, if the likelihood is non-zero at $\lambda = 0$ (respectively, $S = 0$), then when the improper unbounded uniform prior on $S$ ($\lambda$) is used, the posterior will also be improper and unbounded. In practical applications, this problem has generally been masked by the use of an upper bound on





the prior, but (while a lower bound of 0 may be defended on the basis of stability) the choice of upper bound is hard to justify. The upper bound which appears to have been most commonly used for sensitivity is 10°C and we will adopt this choice here. We use a range of $0.37 - 10$ for the uniform priors in both $\lambda$ and $S$, which ensures that their ranges are numerically identical (although their units are of course different). As a third alternative prior for $S$, we will also use the positive half of a Cauchy

prior, with location 0 and scale parameter 5, ie $p(S) = \frac{2}{5\pi(1+(S/5)^2)}$, $S > 0$. An attractive feature of the Cauchy prior is that it has a long tail which only decreases quadratically (hence it does not rule out high vales *a priori*) and moreover, its inverse is also Cauchy so both $S$ and $\lambda$ have broad support. The scale factor is the 50th percentile of the distribution hence the half-Cauchy prior for $S$ has a 50% probability of exceeding 5°C. The scale factor of the corresponding implied prior in $\lambda$ is given by $3.7/5 = 0.74 m^{-2} K^{-1}$.

## 3  Applications

We now consider three areas in which observational constraints have been used to estimate the equilibrium climate sensitivity. Firstly, we consider paleoclimatic evidence, which relates to intervals during which the climate was reasonably stable over a long period of time and significantly different to the pre-industrial state. We then consider analyses of the observations of the warming trend over the 20th Century (strictly, extending into the 21st and 19th century). Finally we consider analyses of

interannual variability.

### 3.1  Paleoclimate

#### 3.1.1  Observationally-derived PDFs

A common paradigm for estimating the equilibrium climate sensitivity $S$ using paleoclimatic data is to consider an interval in which the climate was reasonably stable and significantly different to the present, and analyse proxy data such as pollen grains

and isotopic ratios in sediment cores in order generate estimates of the global mean temperature anomaly $\Delta T$ and forcing anomaly $\Delta F$ relative to the current (pre-industrial) climate. $S$ can then be estimated via the equation

$$S = F_{2\times} \times \Delta T / \Delta F \qquad (3)$$

where $F_{2\times}$ is the forcing due to a doubling of the atmospheric $CO_2$ concentration. Examples of this approach include Annan and Hargreaves (2006) and Rohling et al. (2012).

The interval which has been examined in most detail in this manner is probably the Last Glacial Maximum, 19–23ka (Mix et al., 2001) where the climate was reasonably stable (at least in the sense of gross evaluations such as global mean surface air temperature on millennial time scales) and substantially different to the present day such that the signal to noise ratio in estimates of forcing and temperature change are reasonably high.

The method adopted by Annan and Hargreaves (2006) and we believe many others (although this is not always documented

explicitly), which we term here 'sampling the observational pdfs', was to generate an ensemble of values of $S$ by repeatedly drawing pairs of samples from pdfs which are deemed to represent estimates of the forcing and temperature anomalies, and





calculating for each pair the corresponding value of $S$ using Equation 3. The ensemble of values for $S$ so generated is then considered as a representative sample from a probabilistic estimate of the truth.

Using values based broadly on those used in Annan and Hargreaves (2006); Köhler et al. (2010); Rohling et al. (2012); Annan and Hargreaves (2013), we use here observational estimates of $5 \pm 1.5°$C for $\Delta T$ and $9 \pm 2 W m^{-2}$ for $\Delta F$ (with the observational errors assumed to represent one standard deviation of a Gaussian), along with a fixed value for $F_{2\times}$ of $3.7 W m^{-2}$. In the illustrative calculations presented here we ignore any issues relating to the non constancy of the sensitivity $S$ and how it might vary in relation to the background climate state and nature of the forcing, although we have slightly inflated the uncertainties of the observational constraints in order to make some attempt to compensate for this. Thus the numerical values generated here are not intended to be definitive but are still adequate to illustrate the different approaches.

Although they may not have been clearly presented as such, most published estimates for $\Delta T$ should be correctly understood as representing likelihoods $p(\Delta T_o | \Delta T_T)$ – that is to say, the observational analysis provides an uncertain estimate of the true value of the form given by Equation 1 with an *a priori* unbiased error of the specified value. This follows immediately from a description of the observational evidence, which is typically based on analyses of data from proxies which is calibrated so as to represent temperature estimates with reasonably well-characterised uncertainties. The result of Annan and Hargreaves (2013) certainly fits this description. Adopting such a likelihood as a pdf for the true temperature anomaly is therefore a clear example of the confusion of the inverse fallacy.

For the forcing estimate, things are not so clear. We do not have direct proxy-based evidence for the forcing, which is typically estimated based on a combination of modelling results and some rather subjective judgements (Köhler et al., 2010; Rohling et al., 2012). Any uncertainty in the actual measurements involved, such as those of greenhouse gas concentrations in bubbles in ice cores, makes a negligible contribution to the overall uncertainty in total forcing. Therefore, we do not have a clear measurement model of the form given in Equation 1 with which to define a likelihood for the forcing. Thus we consider the stated distribution to directly represent a prior estimate for the forcing anomaly. We do not claim that this is the only reasonable approach to take here and other researchers might prefer to make different choices, in particular if they could clearly identify a likelihood arising from observational data.

When applied to the numerical estimates provided above, the pdf-sampling method of Annan and Hargreaves (2006) generates an ensemble for $S$ with a median estimate of $2.1°$C and a 5–95% range of $1.0$ to $3.8°$C. Figure 1 presents this result as the cyan line, together with additional results which will be described below.

### 3.1.2 Bayesian interpretation and alternative priors

Now we present alternative calculations which take a more standard explicitly Bayesian approach. We start by writing the model in the form

$$\Delta T = S \times \Delta F / 3.7 \tag{4}$$

or equivalently

$$\Delta T = \Delta F / \lambda \tag{5}$$





where $\lambda = S/3.7$ is the feedback parameter. This formulation allows us to easily consider the forcing and feedback parameter as uncertain inputs to the model, which can then be updated by the likelihood arising from the observed temperature change.

Although the method of sampling observational pdfs described in Section 3.1.1 was not presented in Bayesian terms, we are now in a position to present a Bayesian interpretation of it. The distribution generated by sampling the pdfs is distributed as

independently Gaussian $N(5, 1.5)$ in $\Delta T$ and Gaussian $N(9, 2)$ in $\Delta F$. We aim to generate this as the posterior of our Bayesian analysis. The likelihood for $\Delta T$ as described above is taken to be the Gaussian $N(5, 1.5)$ and therefore (by rearrangement of Bayes' Theorem) the prior must be uniform in $\Delta T$ and independently Gaussian $N(9, 2)$ in $\Delta F$. For numerical reasons we must impose bounds on the uniform prior for $\Delta T$ and set this range to be 0–20°C.

Using Equation 3, we can reparameterise this joint prior distribution over $\Delta T$ and $\Delta F$ into $S$ and $\Delta F$, and this is presented

in Figure 2. Note that this prior cannot be represented as the product of independent distributions over $S$ and $\Delta F$, as high $S$ here is correlated with low $\Delta F$ and vice versa. The prior in $S$ when viewed as a marginal distribution (i.e., after integrating over $\Delta F$) appears uniform over a significant range (roughly between $S = 0.6$ and $S = 5$) but within this range it is associated with somewhat high values for $\Delta F$, with the latter taking a mean value of about $9.5 W m^{-2}$ over this region. The details of the shape of this joint prior does depend on the bounds placed on the uniform prior for $\Delta T$, but this does not affect the posterior

so long as the prior is broad enough to cover the neighbourhood of the observation. We think it is unlikely that researchers would choose a joint prior of this form deliberately, and confirm that this certainly was not the case in Annan and Hargreaves (2006). In future analyses it would seem more appropriate to clearly state the priors which are used, and test the sensitivity of the results to this choice.

In order to perform a more conventional Bayesian updating procedure using Equation 5, we must first select priors on the

model inputs. For the forcing $\Delta F$, we retain the $N(9, 2)$ prior, having no plausible basis for trying anything different. For sensitivity, we test the three priors described in Section 2.2. The two uniform priors generate rather different results. Using a prior which is uniform in $S$ the posterior has a mean value for $S$ of 2.2°C and a 5–95% range of 1.0–4.2°C. When we change to uniform in $\lambda$ the median decreases to 1.5°C with a 5–95% range of 0.5–3.0°C. While these results, which are shown in Figure 1, overlap substantially, broadening the upper bounds on the priors would result in the first result increasing without limit and the

second decreasing towards zero such that they would fully separate. We therefore see that extreme choices for the prior on $S$ (or $\lambda$) can have significant influence on Bayesian estimation, which is perhaps not surprising given the large uncertainties in the observational constraints used here. The median posterior value for $S$ obtained from the half-Cauchy prior is 2.1°C with a 5–95% range of 1.0–3.8°C, which coincidentally aligns very closely with the result obtained by the naive method of sampling observational pdfs (which is plotted as a thick line in Figure 1 in order to make it more visible). We conclude in this case that

the method of sampling pdfs has generated a result which is reasonable but alternative choices of prior could give noticeably different results.



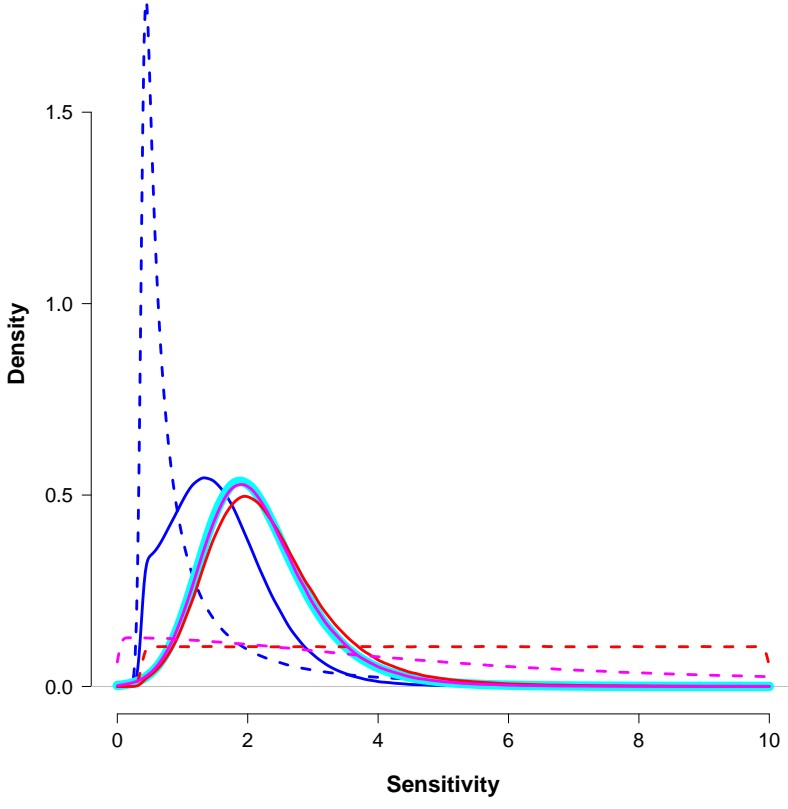

**Figure 1.** Prior and posterior estimates for the climate sensitivity arising from paleoclimatic evidence. Dashed lines show priors, solid lines are posterior densities. Thick cyan line shows posterior estimate arising from the method of sampling observational pdfs, with corresponding prior shown in Figure 2. Blue lines represent results using uniform prior in $\lambda$, red is uniform in $S$, and magenta is half-Cauchy (scale = 5) in $S$ (and therefore also half-Cauchy (scale = 3.7/5) in $\lambda$).



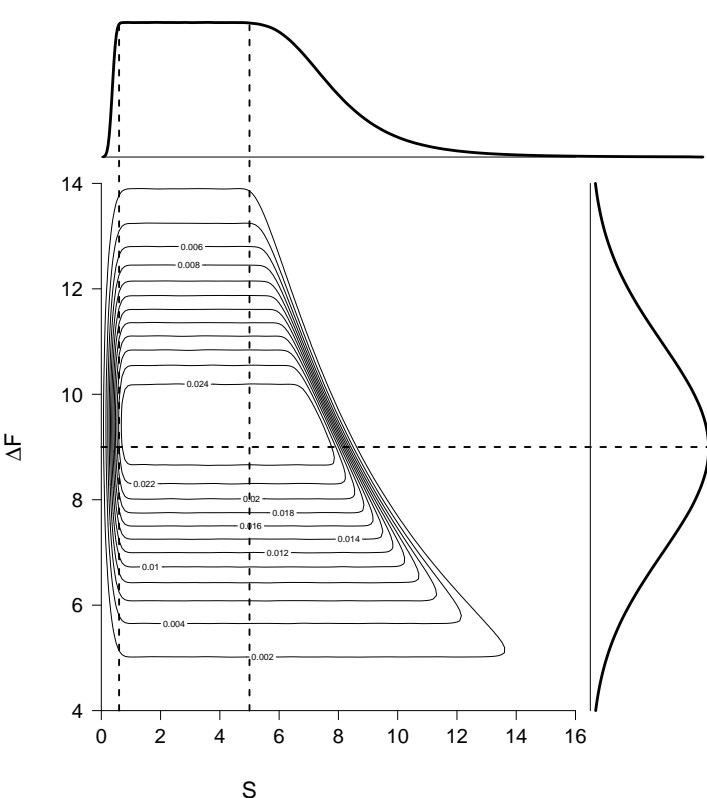

**Figure 2.** Implicit prior used in paleoclimate estimate. Contour plot shows joint prior in $S$ and $\Delta F$ with marginal densities shown at top and right respectively. Vertical and horizontal dashed lines drawn at $S = 0.6$, $5$ and $\Delta F = 9$.

## 3.2 Estimates based on historical warming

### 3.2.1 Observationally-derived PDFs

Perhaps the most common approach to estimating $S$ has been to use the instrumental record (Tol and De Vos, 1998; Gregory et al., 2002; Olson et al., 2012; Aldrin et al., 2012). While a wide range of climate models have been utilised for this purpose, a

5 simple energy balance similar to that of Section 3.1 can be used so long as the radiative imbalance is accounted for. We follow the recent analysis of Mauritsen and Pincus (2017) but simplify their calculation by ignoring uncertainty in $F_{2\times}$, instead adopting their mean value of $3.71 W m^{-2}$ (using all their uncertain numerical values otherwise). This simplification has very little influence on the results. Mauritsen and Pincus (2017) present the basic energy balance in the form

$$S = F_{2\times}\Delta T/(\Delta F - \Delta Q) \tag{6}$$





where $\Delta Q$ represents the net planetary radiative imbalance and the other terms are as before. This equation is applied between two widely separated decadal-scale intervals within the historical record, such that the signal to noise ratio (and hence precision in the resulting estimate of $S$) is as large as possible, though it remains a significant source of uncertainty (Dessler et al., 2018). Similar to Section 3.1.1, the method used by Mauritsen and Pincus (2017) is one of sampling observationally-derived pdfs for

all uncertain quantities on the right hand side of equation 6, and thereby generating an ensemble of values for $S$ which was interpreted as a probability distribution.

### 3.2.2 Bayesian interpretation and alternative priors

Reorganising the equation in order to give $\Delta T$ as the prognostic variable we obtain

$$\Delta T = (\Delta F - \Delta Q) \times S/F_{2\times} = (\Delta F - \Delta Q)/\lambda \tag{7}$$

As before, we adopt the distributions used by Mauritsen and Pincus (2017) for $\Delta F$ and $\Delta Q$ as priors for these variables, but recognise their estimate for $\Delta T_o \sim N(0.77, 0.08)$ as a likelihood $p(\Delta T_o|\Delta T)$ arising from the measurement model of Equation 1. Given the similarities between Equations 3 and 6, and also in the method used, it is no surprise to find that the implicit prior used here before updating with the temperature likelihood is qualitatively similar to that found in Section 3.1. This is shown in Figure 3. Again the marginal prior over $S$ appears uniform over a reasonable range (the details depend on the

limits of the uniform prior over $\Delta T$) but nevertheless it is actually correlated with the net forcing. Figure 4 shows the posterior result arising from this prior, which matches the published result of Mauritsen and Pincus (2017) closely despite our minor simplification to their calculation. The posterior median calculated here is 1.8°C with a 5–95% range of 1.1–4.5°C. As in Section 3.1, we make no attempt to decompose the forcing estimate into prior and likelihood, especially as some of the largest uncertainties (e.g. that arising from aerosol forcing) are based on modelling calculations and expert judgments that cannot be

transparently traced to uncertainties in observational data.

Alternative priors and their resulting posteriors after Bayesian updating using Equation 7 are shown in Figure 4. As before, we test the three priors presented in Section 2.2. The posterior median (and 5–95% range) for $S$ arising from these are 2.1°C (1.2–6.3°C) for uniform-$S$, 1.5°C (1.0–3.1°C) for uniform-$\lambda$ and 2.0°C (1.1–5.0°C) for the half-Cauchy prior respectively. Thus again the half-Cauchy prior produces a result which is intermediate between the other explicit choices, though this time it

has a somewhat longer tail than the pdf sampling method. The differences between these results, especially for the upper 95% limit, are substantial and could significantly alter their interpretation and impact.

### 3.3 Estimates based on interannual variability

### 3.3.1 Observationally-derived PDFs

Finally, we consider a method which has been used to estimate the climate sensitivity via interannual variation in radiation

balance and temperature (Forster and Gregory, 2006; Dessler and Forster, 2018). The basic premise of these analyses is that the feedback parameter can be estimated as the slope of the regression line of net radiation imbalance (based primarily on satellite

Joint and marginal densities for S and ΔF − ΔQ

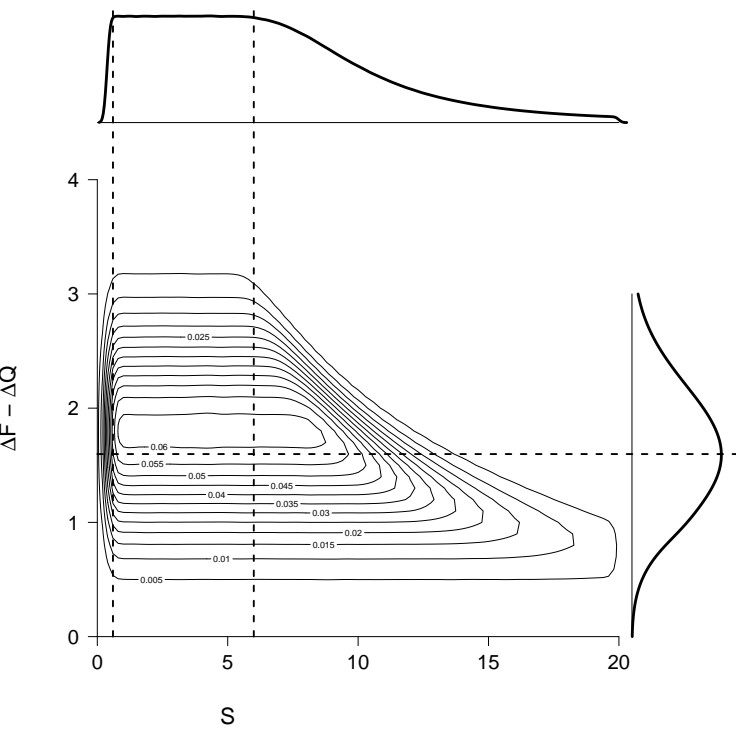

**Figure 3.** Implicit prior used in 20th century estimate.

observations) against temperature anomalies, with data typically averaged on an annual time scale (though seasonal data may also be used). There are questions as to whether this short-term variability provides an accurate estimate of long-term changes, but this is beyond the scope of this manuscript (Dessler and Forster, 2018). The regression slope and its uncertainty naturally translates into a Gaussian likelihood for the true feedback component, and has been commonly interpreted as a probability

5  distribution for $\lambda$. While this again appears to commit the fallacy of confusion of the inverse, the implicit assumption of a uniform prior on $\lambda$ has been clearly acknowledged by authors working in this area (e.g. see comments in Forster and Gregory (2006) and Forster (2016)). In this section we will use the observational estimate of Forster and Gregory (2006) which is given by $\lambda_o = 2.3 \pm 0.7 W K^{-1}$.



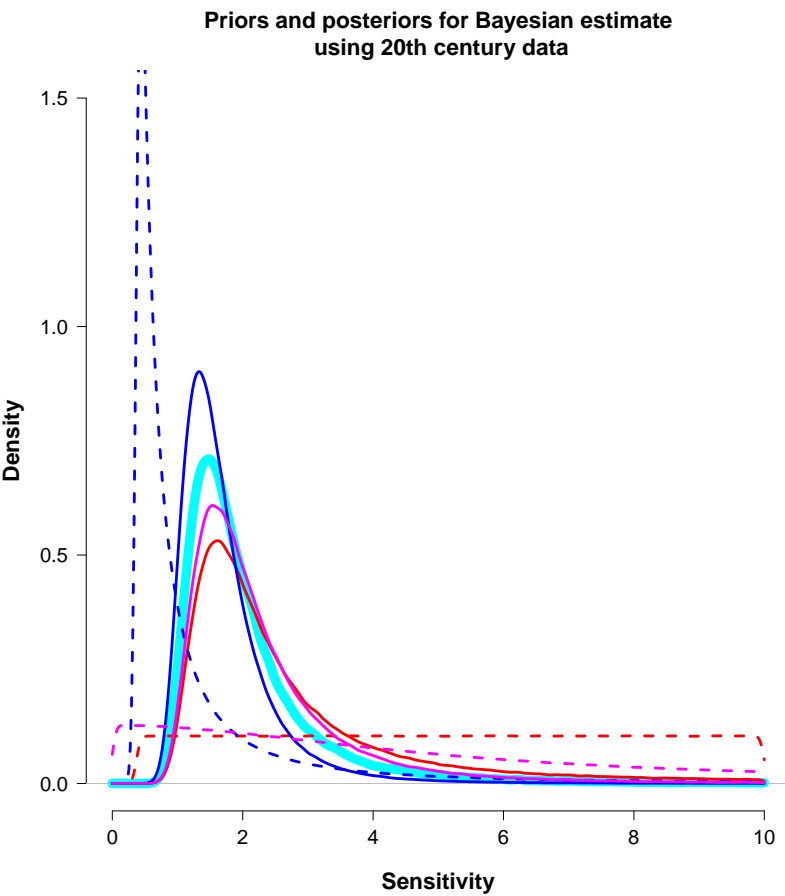

**Figure 4.** Priors and posteriors in explicit Bayesian estimates using 20th century data. Dashed lines show priors, solid lines are posterior densities. Thick cyan line shows posterior estimate arising from the method of sampling observational pdfs, with its implicit prior shown in Figure 3. Blue lines represent results using uniform prior in $\lambda$, red is uniform in $S$, and magenta is half-Cauchy (scale = 5) in $S$ (and therefore also half-Cauchy (scale = 3.7/5) in $\lambda$).

**Figure 5.** Priors and posteriors over $S$ in process-based feedback analysis. Dashed lines indicate priors, solid lines are posteriors. Thick cyan line shows shows posterior estimate arising from the method of sampling observational pdfs, which coincides precisely with the blue line which corresponds to the uniform prior in $\lambda$. Red lines show results using uniform prior in $S$ and magenta is half-Cauchy (scale = 5) in $S$.





### 3.3.2 Bayesian interpretation and alternative priors

As noted by Forster and Gregory (2006), presenting what actually amounts to an observational likelihood for $\lambda$ as a posterior pdf is equivalent to assuming a uniform prior in $\lambda$. Therefore the Bayesian interpretation is already clear in this instance.

In Figure 5 we present the results of calculations using our three alternative priors (albeit one of them coincides with the method of sampling pdfs). The original result of Forster and Gregory (2006) (after transforming to $S$-space) is represented by the blue lines, with red showing the result obtained for a uniform prior in $S$ and magenta being a Cauchy prior. We note that, for the uniform-$S$ case, if the upper bound on the prior was raised, the posterior would also increase without limit due to the pathological behaviour discussed in Section 3.1.2 and also by Annan and Hargreaves (2011). For the priors shown (with the uniform priors defined as $U[0.37, 10]$) the 5–95% ranges of the posteriors are 1.1–3.2°C, 1.2–6.9°C and 1.2–5.2°C for the uniform-$\lambda$, uniform-$S$ and Cauchy-$S$ priors respectively. The uniform-$\lambda$ prior commonly adopted by analyses of this type provides a strong tendency towards low values and the the contrast with uniform-$S$, especially for the upper bound, is disconcerting.

## 4 Conclusions

We have shown how various calculations which have presented probabilistic estimates of the equilibrium climate sensitivity $S$ can be reinterpreted within a standard Bayesian framework. Using this standard framework ensures a clear distinction between the prior choices which must be made for model parameters and inputs, and the likelihood obtained from observations of the system which is then used to update this prior in order to generate the posterior.

In many cases, the implied prior for $S$ which underlies the published results appears somewhat unnatural, having either a structural relationship with model inputs or a marginal distribution that may not be considered reasonable. We have presented alternative calculations in which a range of simple priors are tested. As well as the commonly-used uniform priors, we have shown that a Cauchy prior has some attractive features, in that it extend to high values (refuting any suspicion that the results obtained were simply constrained by the prior), its reciprocal is also Cauchy (so both $S$ and $\lambda$ may have long tails). The half-Cauchy distribution used in this paper only requires a single scale parameter which determines the width. However the choice of priors is always subjective and we make no assertion that this choice should be universally adopted. Indeed there may be superior alternative choices that we have not considered.

Our calculations suggest that the pdf sampling method can generate acceptable results in some cases, agreeing fairly well with a fully Bayesian approach using reasonable priors. However, this is not always the case. We recommend that researchers should present their analysis in an explicitly Bayesian manner as we have done here, as this allows the influence of the prior and other uncertain inputs to be transparently tested.

*Author contributions.* Both authors contributed to the research and writing.





*Acknowledgements.* We are grateful to Andrew Dessler for helpful comments on the manuscript.





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
