# Peer review of "Bayesian deconstruction of climate sensitivity estimates using simple models: implicit priors, and the confusion of the inverse."

_Earth System Dynamics, 2019_

## Referee Comment (RC1) · Anonymous Referee #1 · 9 Aug 2019

Review of Annan and Hargreaves, "Bayesian deconstruction of climate sensitivity estimates using simple models: implicit priors, and the confusion of the inverse"

This paper is a tutorial arguing that researchers replace PDF sampling with a Bayesian framework. Overall, I quite liked the paper. I thought it was written at the right level and was very accessible to a non-expert like myself.

I recommend publishing this after the authors consider my comments below.

I have only one comment that I would strongly suggest the authors accept: they should put some code on-line that shows how they did the calculations. The advantage to doing this is that it will show the details of the calculation in a detail that it is not possible to

[Figure]

put into words in the manuscript. Overall, I was able to follow generally the explanation in the text, but I think I would have trouble actually coding up the procedure. Seeing example code would be very very helpful.

Ideally, they would include all of the code needed to generate the figures, but if they don't want to do that they could put the code on-line for one of the examples in Sect. 3.1, 3.2, or 3.3. There are lots of free places to put code (e.g., zenodo.com or, of course, github) that work well.

More minor comments: The abstract needs to be rewritten. It reads like the introduction of the paper, and it has paragraph breaks within it. I would try to make it more of a summary of the main points of the paper.

Last line on page 2: where does the 0.5 come from?

In equation 2, what part of that is the "prior"? in fact, the term "prior" doesn't seem to be defined anywhere in the paper, which seems to me to be an oversight.

Top of page 8: I would add a sentence here making explicit what you're doing: you're trying to back out what prior you'd need to get the same answer from a Bayesian analysis as you do from the naïve PDF sampling.

In e.g., Eq. 7, the authors re-arrange the equation so that delta T is on the LHS. Why is that done? It seems important, but I'm a bit lost.

Sect. 3.3.1: I am quite confused what's going on here. I think what they're doing is taking lambda from Forster and Gregory and then using a Bayesian analysis to convert that to a value of S. Is that right? I think that they could add just a few words to make this more clear.

---

## Referee Comment (RC2) · Anonymous Referee #2 · 19 Aug 2019

**Review: Bayesian deconstruction of climate sensitivity estimates**

This very well written paper attempts to argue for a more transparent approach to quantifying uncertainty in climate sensitivity estimation by using explicit statistical models and justified priors within Bayesian analyses. I agree entirely with this position and so there are things that I like about this paper. However, I believe that the core underpinning assumption that the authors make, namely that interval estimates given by researchers on observations should be treated as likelihoods within a Bayesian framework, cannot be justifiably imposed on a researcher and makes little sense even if it could be. I will detail my objections below and suggest a correction that would enable calculations to be redone and the paper to be resubmitted. I am therefore recommending that the paper be rejected and, if the authors wish to make alterations to their methods and argument, resubmitted for publication. I want to stress that I like the concept of the paper and the idea to examine climate sensitivity estimates as if certain priors were chosen is a nice one, so I genuinely would hope that the authors do resubmit having reflected on the comments below.

**Major corrections**

The critical problem in the paper's argument is as follows: Assuming, using the notation of the paper, that $x_T$ is the true temperature and $x_o$ is an observed temperature, then the measurement model is, as given,

$$x_o \mid x_T \sim \mathrm{N}(0, \sigma^2).$$

The likelihood of any particular measurement, $x_o$ is then $p(x_o \mid x_T)$ and has the form given on page 3 line 20. The likelihood is a function of the unknown and unobserved $x_T$. On line 10 of page 7, the authors make the assumption that is key to the paper's methodological argument, that published estimates for $\Delta T$, should be correctly understood as representing likelihoods $p(\Delta T_o \mid \Delta T_T)$. This cannot be true as is, because the likelihood is a function of the unknown $\Delta T_T$, and an estimate is a given interval. However, the authors clarify their meaning in the second half of the sentence as meaning that this is a 'likelihood' with "an a priori unbiased error of the specified value", which I take to mean that they assume $x_T$ was given an unbiased estimator (not Bayesian) and then plugged into the likelihood. But the algebra doesnt match the description. Line 11 says that $p(\Delta T_o \mid \Delta T_T)$ "provides an uncertain estimate of the true value...", but the true value of $\Delta_T$ is considered known in the given conditional.

The described procedure does not define a likelihood. You might call it the likelihood function at $\Delta T_T = \Delta T_o$, but the authors want to do that in order to claim that researchers have simply adopted a likelihood and then committed the fallacy of the inverse. It is not clear why you should impose that constraint on researchers who have given an interval estimate, particularly when a more natural Bayesian interpretation of an interval is available (see below). Before explaining, it's worth examining the author's claim in the context of their example on page 11. Line 11 states that "we recognise their estimate $\Delta T_o \sim \mathrm{N}(0.77, 0.08)$ as a likelihood $p(\Delta T_o \mid \Delta T)$...". So, the idea is to take a researcher's distribution for $\Delta T_o$, given unconditionally, and assume that it's derivation has committed a fallacy because they really supposed that they knew $\Delta T$, the key unknown they'd like to estimate, and fixed it at $\Delta T_o$. The authors are correct to point out that this would be problematic, but it is not a natural interpretation.

When we see a distribution such as $\Delta T_o \sim \mathrm{N}(0.77, 0.08)$ or an interval estimate $\Delta T_o \pm \sigma$, the authors are right that we cannot simply suppose that the same distribution holds for $\Delta T$, but that is not to say that we can impose implicit conditioning on $\Delta T$ either. We have a distribution for $\Delta T_o$ and nothing else, and I don't see why we should be able to argue to rewrite the claims of other scientisits to fit a given narrative. A more natural Bayesian interpretation and one that does not assume that "scientists who wrote X really meant Y", is to take the distributions given at face value, so suppose we have been given the distribution for $\Delta T_o$ that

the researchers believe. This is the evidence or the marginal likelihood in the Bayesian paradigm:

$$p(\Delta T_o) = \int p(\Delta T_o \mid \Delta T) p(\Delta T) d\Delta T,$$

where the likelihood given by the usual measurement model is integrated over the prior for $\Delta T$.

As this preserves the given intervals and still enables the authors to take their approach in investigating implicit priors and changing them to more natural ones, a revised version of the paper should do this. It may be that the description of the implicit conditioning resonates with certain scientists who publish interval estimates, so the authors could keep some of their originial analysis, but more carefully caveated.

But even the above correction, still assumes that the original researchers were subjective Bayesians (or at least ascribed to a subjective interpretation of probability). A difficulty I have with interpretations of analyses from one philosophical school (e.g. frequentism) in terms of the others (e.g. subjective Bayes), is that they are fundamentally incompatible and a great deal of straw manning the other side is required to obtain a coherent argument. I am a subjective Bayesian, and I would argue that Bayesian approaches are the only ones that make sense in any given situation. However, if a researcher has undertaken a frequentist analysis, I would be very careful not to overinterpret. A key example here is alluded to on line 9 of page 5. If uncertainties in measurements are purely random errors coming from a measuring device, the $\sigma$ in the measurement model has a clear frequentist meaning (up to the debate about what purely random means, wherein Bayesians and Frequentists are likely to also disagree). A frequentist analysis can admit no further type of uncertainty here, nor would they mean to when they reported a confidence interval for $x_T$. It doesnt even make sense to talk about a probability distribution for $x_T$, as $x_T$ is not random, and a good frequentist would not make the mistake that it was. A Bayesian might also consider contributions to uncertainty through assumptions required to compute the derived observed quantity and other things that are unknown, but not random. The very interpretation of the measurement model and what is/can be included in $\sigma$ can be different, and it makes sense to talk about a probability for $x_T$, albeit a subjective one.

The fundamental problem in a lot of the mentioned publications is that the statistical model is not clearly written down, the assumptions are not clearly stated and the meaning of uncertainty statements is not clearly given. This leaves a void in which readers are free to project their own definitions of "uncertainty" and where the meaning of inferred distributions or intervals is open to interpretation. The authors here have recognised this and attempt to give some interpretation, but there is a danger that with any of the analyses they picked (perhaps with the exception of their own), that they are straw manning the position of the authors. A safer way to do what the authors want to do is to make statements such as "if paper X meant Y by this statement, then that would mean Z, but A, B and C are also possible interpretations". It's harder work of course, but, that is the problem with traversing the different philosophical positions, or, more precisely, with attempting to ascribe subjective Bayesian interpretations to existing analyses that are unlikely to have been conducted by researchers who hold those views. Aside from frequentism, there are also at least 3 kinds of Bayesian (subjective, objective, as the authors mention, and falseificationist), and each would mean a different thing by the modelling statements and conclusions that were made. I like the Bayesian approach that is taken in the paper, and I see no reason not to advocate for it. I do believe that the set up and the discussion and interpretation of other work should bear the above in mind.

My final point is on the first 4 pages where Bayesian statistics and the prosecutors fallacy are exaplined in far too much detail. Bayes is widely used in climate science, so I think a derivation of Bayes theorem (seen in any first year undergraduate probability/statistics course) is overly indulgent. Similarly, the prosecutor's fallacy is well known and is taught in every entry level probability and statistics course, so a page and a half of a research article should not be given over to explaining it in detail with examples. A quick outline and references should be enough.

---

## Author Comment (AC1) · 12 Sep 2019

Thank you for your comments. Our specific responses are given below.

Major comment:

Regarding code availability: yes we will make code available. Probably the easiest solution would be to provide the (commented) code that generates the figures. This code is simple and short (written in the language R with few external packages) and hopefully can accompany the paper as supplementary information rather than in a separate repository. We will consult with the Editor concerning this.

[Figure]

Minor comments.

The abstract will be rewritten.

*Last line on page 2: where does the 0.5 come from?*

0.5 is twice the standard deviation of observational error in this example (0.25) which will be made clearer.

*In equation 2, what part of that is the "prior"? in fact, the term "prior" doesn't seem to be defined anywhere in the paper, which seems to me to be an oversight.*

$p(x_T)$ is the prior which will be made explicit

*Top of page 8: I would add a sentence here making explicit what you're doing: you're trying to back out what prior you'd need to get the same answer from a Bayesian analysis as you do from the naïve PDF sampling.*

Agreed

*In e.g., Eq. 7, the authors re-arrange the equation so that delta T is on the LHS. Why is that done? It seems important, but I'm a bit lost.*

Using this arrangement, all terms on the RHS are already known/defined (in distribution) so we can conveniently generate the corresponding prior predictive distribution for $\Delta T$ and use Bayes' Theorem to update this using the likelihood for $\Delta T$.

*Sect. 3.3.1: I am quite confused what's going on here. I think what they're doing is taking lambda from Forster and Gregory and then using a Bayesian analysis to convert that to a value of S. Is that right? I think that they could add just a few words to make this more clear.*

Yes, agreed
* * *

---

## Author Comment (AC2) · 13 Sep 2019

Thank you for the interesting comments.

There appear to be several distinct issues raised, namely:

(a) definition of likelihood (as discussed on most of p1–2 of the review)
(b) straw-manning (p2)
(c) excessive/indulgent detail (last paragraph p2)

Addressing these in reverse order:

(c) We wish we could agree with the reviewer on this point, but this introduction was

recently expanded at the specific request of a reader who found a previous shorter version of the manuscript rather too terse. Reviewer #1 has also asked for some additional explanation. It may seem a rather unsatisfactory state of affairs but the fact is that these sort of calculations are routinely carried out by a wide range of researchers who are not going to go away and take undergraduate statistics classes or read statistics textbooks however desirable this would be. We have presented evidence that misunderstandings associated with the confusion of the inverse are widespread, perhaps even ubiquitous, and it is not uncommon for undergraduate-level teaching material to be misleading on these issues. See eg p116 of the first edition of Wilks 'Statistical methods in the Atmospheric Sciences' where it is said of classical frequentist hypothesis testing: "If the test statistic falls in a sufficiently improbable region of the null distribution, $H_0$ is rejected as too unlikely to have been true given the observed evidence". As another example, the STEPS glossary at http://www.stats.gla.ac.uk/steps/glossary/confidence_intervals.html says "A confidence interval gives an estimated range of values which is likely to include an unknown population parameter". Since statistical authorities can make such confusing statements we hope we can be indulged with a bit of extra commentary explaining why this is wrong. In our experience, a large majority of scientists (including ourselves) have often misinterpreted frequentist confidence intervals in this way, and it takes quite a bit of time and care to explain why this interpretation is invalid.

(b) On the issue of straw-manning, we broadly agree with the reviewer that the manuscript would read better if we were less dogmatic in imputing motive and/or belief to authors. To that end, we agree that the paper would be better edited along the lines of the reviewer's "if paper X meant Y …" construction and plan to revise it throughout along these lines. However, in our defence, we should mention that our manuscript was originally motivated by discussions with some of the cited authors who had observed the discrepancy arising from the two calculation methods using otherwise identical numerical values and wondered if there was a way of reconciling the two approaches and/or a route to interpreting the 'sampling the pdfs' method within

the Bayesian paradigm. It is very clear from our discussions with them, that they had indeed been working under the assumptions as we have stated, namely that an uncertain observation can be assumed to be a probabilistic estimate for the measurand in the way described. Moreover, this manuscript has also been widely circulated to a wider range of relevant researchers (including several more of those cited) for their views and none of them complained that we were incorrectly putting words into their mouths. Nevertheless, we acknowledge that it would be better written in a more neutral manner.

(a) The most fundamental criticism appears to be:

*However, I believe that the core underpinning assumption that the authors make, namely that interval estimates given by researchers on observations should be treated as likelihoods within a Bayesian framework, cannot be justifiably imposed on a researcher and makes little sense even if it could be.*

The reviewer appears to endorse our introduction using the general notation of $x_T$ and $x_o$ in the measurement model, up to and including the resulting definition of likelihood on line 20 of page 3. They then criticise our interpretation of temperature estimates $\Delta T$. Our intention was for this interpretation to be mathematically identical to that of $x$, merely making the formal substitution of notation in order to pass from the general to the specific case. Therefore, we are unsure what to make of the reviewer's comment "This cannot be true as is, because the likelihood is a function of the unknown $\Delta T_T$, and an estimate is a given interval." When researchers present an uncertain estimate in interval form $\mu \pm \sigma$, in the vast majority of cases they don't simply mean to convey an interval $[\mu - \sigma, \mu + \sigma]$ and nothing more, but almost invariably have a measurement model similar to that of equation 1 in mind and are merely using the interval as a convenient notation to summarise the observed value and magnitude of its uncertainty. This can be seen very clearly in Mauritsen and Pincus (2017) where this interval notation (in their case presented as as 5–95% intervals) is used widely to represent Gaussian distributions. We don't believe this is at all unusual or controversial but will happily

note it as an explicit assumption of our analyses. We also admit the existence of some observations which do not fit to our simple model, one example of this could be where the magnitude of observational uncertainty is not assumed to take a constant known value but instead varies with $x_T$. However we do not believe that this is relevant to the observations discussed in this manuscript.

Therefore, while we do agree that we cannot automatically impose this type of likelihood on researchers in all cases, we don't understand the reviewer's comment (eg in their first paragraph) that "it makes little sense to do so", at least in the specific situations we have described.

A typographical error in our manuscript may have helped to cause confusion. Where we wrote $\Delta T_o \sim N(0.77, 0.08)$ on p11 this would better have been written as $\Delta T_T \sim N(0.77, 0.08)$ or otherwise reworded in such a way as to make the meaning clearer. $\Delta T_o$ is simply a fixed known value (once the observation has been made) and has no non-trivial distribution. Thus the reviewer's comments referring to a distribution for $\Delta T_o$ have no applicability. There is no such distribution (at least once the observation has been made).

In summary, we would like to submit a revised version of the manuscript in which we make clear that the analyses presented here depend on the assumption of a measurement model of the type presented in Equation 1. Moreover, we will try to avoid any imputation that cited authors have incorrectly interpreted the observational analyses, but rather outline how different interpretations could arise and demonstrate (as we have shown) how they can be reconciled within a Bayesian framework with particular priors.

---

## Author Response (AR1)

**Summary of changes**

J. D. Annan & J. C. Hargreaves

October 24, 2019

In relation to Referee #1

Major comment:

Code is available to generate all figures, which should help to clarify how the calculations proceed in practice.

Minor comments:

We believe we have improved the abstract and more directly summarise the content of the manuscript.

last line of p2 a clear reference is now made to the previously mentioned standard deviation of 0.25

Prior is defined in Equation 2

p8: additional explanation of inversion procedure added.

Eqn 7: We realised during revision that this is equivalent to the rearrangement in Eqns 3-5 so we refer back to the explanation provided there.

3.3.1 Added a couple of sentences at the end of this section to make explicit the 1–1 equivalence between $\lambda$ and $S$.

In relation to Referee #2

In regard to point (c) from our original response (excessive detail), we have no doubt that there is a gulf between the level of detail that many scientists performing these calculations would like to see (and in some cases have specifically asked for), and that which can be honestly justified as novel research. We are certainly biasing far more towards the former as a deliberate choice due in large part to our recent experiences collaborating with scientists working on these topics, and also in view of our intended audience who we do not expect to be statisticians.

Relating to what we previously described as points (b) and (a) together, as they appear to be rather similar (broadly speaking, the validity of imposing a particular statistical framework on the published research), we have added some additional language to clarify that our interpretations and reanalyses are somewhat personal and could possibly be considered in some cases to be straw-manning others' work. See in particular the end of Section 2.1, and some more minor changes are also made in Section 3.2.2.

However we would also like to defend our original presentation a bit more robustly than in our previous reply. Specifically, addressing the analyses in reverse order within the manuscript: Forster and Gregory explicitly draw attention (in more than one of their papers) to the implicit prior which can be assumed to have been used their analyses, which directly supports our presentation. That is, their own presentation implies that the observational Gaussian functional form that they use for $\lambda$ is in principle a likelihood and interpreting it as a pdf is equivalent to using it to update a uniform prior. This interpretation was also previously described by Annan and Hargreaves 2011. For the historical section of our manuscript, the forced temperature change of Mauritsen and Pincus is (via the cited Lewis and Curry) seen to arise from a measurement model of the form in Equation 1, where in this case the true value of forced warming is imperfectly observed due to internal variability which can reasonably be assumed unbiased and independent of the forced response. Furthermore, likelihoods of this form have

regularly been used in explicitly Bayesian analyses of this type such as the cited work by both Aldrin and Olson. Therefore we really don't think it is in any doubt that the numerical values of observations of temperature anomalies are most naturally interpreted as likelihoods in the manner described. In the context of paleoclimate temperature estimates, these again generally arise though some underlying model of a somewhat smooth temperature anomaly field which is imperfectly observed through limited spatial sampling combined with uncertainties on the values even where observations do exist (eg as in Annan and Hargreaves 2013). It does not seem controversial that the estimate arising from such an analysis, together with its estimated uncertainty, will naturally fit the paradigm of our equation 1. Nevertheless, the caveat at the end of Section 2.1 is acknowledged.

Specifically in relation to the reviewer's comments about the distribution for the observations, we have corrected some ambiguity (error) in the language describing the presentation of observational data in Section 3.2.2. We hope this addresses that issue satisfactorily.

[revised manuscript text omitted]

---

## Author Response (AR2)

Comment 1. We certainly realised that Reviewer #2 didn't like the example, but in contrast Reviewer #1 explicitly praised the accessibility of the paper. Our manuscript isn't very long and electronic journals don't intrinsically have bounds on space. We would certainly prefer to include the example though it's fundamentally a matter of editorial guidelines as to whether it is appropriate. We have cited appropriate literature that demonstrates how widespread the misunderstanding is.

Comment 2. We do agree with the reviewer that it is not _always_ appropriate to interpret an uncertain estimate as measurement with Gaussian error along the lines of our Equation 1. Indeed, we specifically make this point in reference to the estimates of forcing that are used in the paleoclimate and 20th century calculations. The reviewer's further comments here have been helpful in clarifying their objections and we have edited the paper in several places (documented below) to make our intentions more clear on this point.

However, we do firmly believe that our approach is entirely appropriate in the cases that we consider, for the reasons discussed in the paper and in our previous reply to them. It is simply not factually correct of the reviewer to claim that researchers have reported an error bar and nothing more for these estimates that we consider.

Specifically, the reviewer hasn't actually explained why they consider it incorrect to interpret the estimate of forced warming in the 20th century (0.77±0.08) as arising from Equation 1. We emphasise that the value of 0.77 is the (deterministic) value obtained from the temperature record (albeit based on a complex processing of many thousands of daily thermometer readings), whereas the quoted uncertainty of 0.08 is an independently-derived estimate of internal variability of the climate system (at one standard deviation of a Gaussian) on the appropriate time scale - with this uncertainty assumed by the cited authors to dominate other sources of error.

Observational estimates of this nature have regularly been used to derive likelihoods in Bayesian analyses in exactly the manner we propose so we find it hard to believe that the reviewer really means to reject this approach so broadly. The fact that the estimate of the standard deviation of internal variability (and indeed the use of a Gaussian distribution) is somewhat subjective and relies on a combination of model calculations and expert judgement does not contradict or undermine this interpretation in any way. The critical points are that this internal variability is independent of the forced response, and the observational analysis only measures the sum of these two terms. Equation 1 follows as a direct consequence. We have clarified that the "observational error" in Equation 1 is not necessarily a simple instrumental or sampling error but may include any source of discrepancy between the numerical value generated from the observational analysis, and the measurand that the researcher is interested in.

We note that while our manuscript was in review, another paper (Williamson and Sansom 2019) has been published in which two statisticians with substantial experience in climate science have used the same measurement equation (their equation 6) to interpret a published uncertain estimate of a climate parameter as giving rise to a likelihood with which they update a prior - with a uniform prior in the measurand being presented as the "reference prior" that replicates the original published result. The parameter they consider here is a complex function of temperatures and the true meaning of the uncertainty seems if anything a little less clear than in our examples, being calculated as the standard deviation of a series of highly correlated values obtained from multiple overlapping data sets. Nevertheless they simply assert that the uncertainty in the observation can be interpreted in this form and proceed accordingly. There is also the case of the IPCC (2007) reinterpreting a previously published result (which was generated from 20th century warming via the "sampling the pdfs" method

as described in our manuscript) in order to use a uniform prior for sensitivity. See their Box 10.2 Figure 1 and the accompanying caption. Since the paper's first author (Jonathan Gregory) was also an author on this IPCC chapter, it seems that he endorsed this reinterpretation of his previous research, though we have not discussed this directly with him.

In response to comment 2, we have made the following changes to the manuscript relating to these points (page and line numbering refers to the diff file):

p2 l20
explaining the application of the measurement equation in clearer terms, that the "noise" may include a diverse range of uncertainties beyond those arising from pure instrumental error.

p7 l30
emphasising the relevance of the measurement model to Annan and Hargreaves 2013 wherein the temperature estimate was generated in this form

p11 l10 and l19
explaining more fully how the measurement equation relates directly to the temperature change estimate of Mauritsen and Pincus

As for the analysis of interannual variability, since Forster and Gregory already specifically invoked a uniform prior in lambda to justify their interpretation of their observational estimate as a pdf, the likelihood implied in their research is surely uncontentious in this case.

A few additional edits have been made in order to reduce the chance of misinterpretation, such as

p5 l33 and p14 l30
Emphasis that the interpretation presented in this manuscript is our own and making no assertion of

universal applicability or truth.

Comment 3: We disagree that different likelihood interpretations are "just as arbitrary and subjective" as the prior. In terms of the previously published work that we are reanalysing, the model that gives rise to the likelihood has generally been specified in some detail and has significant theoretical and practical basis, whereas the prior over the sensitivity parameter does not. While in an ideal world a Bayesian may present all assumptions on an equal standing at the outset, in reality the models are typically off-the-shelf choices that have been developed and tested in a variety of contexts outside of Bayesian analysis.

We aren't really sure what to make of suggestions that other probabilistic interpretations are possible. Our intention was to present a way of reconciling different calculations that have been made in a particular area of climate science, and interpret them in a simple Bayesian framework that we believe will be useful to others. Nothing in our work prevents other researchers from presenting different approaches if they choose but it seems outside the scope of this work. We have made additional edits to emphasise this more clearly (p5 l33 and p14 l30). Note that we had already stated in the previous revision (p6 l4): "Thus, while we confidently believe our interpretation to be natural and appropriate in most cases, we do not claim it to be universally applicable."

Comment 4. We agree that the Cauchy prior has weaknesses, but are unaware of any prior that does not. As well as exploring the sensitivity of the results to the prior (with three or four different priors tested in each example) we explicitly suggest in the manuscript that researchers make their own choices. There is long and tedious history behind our particular choice of prior (which was picked deliberately to have very broad tails) and while we could explore an even larger range of priors in more detail, it would seem a distraction from the main

purpose of the paper which is primarily to explain how existing analyses can be interpreted and presented in terms of a simple Bayesian framework.

Williamson, D. B., & Sansom, P. G. (2019). How are emergent constraints quantifying uncertainty and what do they leave behind? Bulletin of the American Meteorological Society, BAMS-D-19-0131.1-45. http://doi.org/10.1175/BAMS-D-19-0131.1

Reviewer comments:

 Section 2. I commented on this as a major correction and the authors have not changed it and merely defended the choice to give an undergraduate statistics lesson in a peer reviewed journal article. If climate statistics is as bad as they claim in their defence then that is a sad state of affairs. However, it is my experience that many climate journals publish proper statistical analyses using Bayes or even more advanced theory. Examples can be found in the works of Berliner, Guttorp, Wilkinson and many more. If this paper must be published the reader's time should not be wasted in the main text with this and it should go in the appendix. It is worth adding that the objection that everyone is misinterpreting things is quite funny given that the basic stats class everyone who uses confidence intervals must have taken at some point would have gone to great lengths to explain why they are not probabilities during that introduction.
    The "natural" approach to assume DT = \mu ± \sigma means that the true value is \mu measured with Gaussian random noise is not natural in general. I am not convinced by their handful of examples. I could spend a long time ranting about this, but to be brief: imposing infinitely many probability statements on a researcher that has reported an error bar is not justifiable, even if you really really want it to be. You can of course decide to proceed in a world where we imagine for the

sake of argument that this is what is implied and go from there. But to say this is normally what is meant can't be justified. When a numerical analyst reports an estimate with error, they very much understand that they are not invoking a measure. They might call it uncertainty, but they are talking about a bound on a numerical error (that can be derived due to the properties of the procedure etc). It is my experience that errors in observational products are very often dominated not by measurement error (which lends itself to the measurement error model in the paper if an author should choose), but by errors in assumptions and the methods used to bring the product together. The 1Sv error in the Rapid array observations for MOC is like this. Tiny observation error, but aggregating into a product requires assumptions which induce one and that error is not computed probabilistically. Global temperature must be similar. Sure, we might justify any observation having a Gaussian error, but a product?? Why? There was an opportunity to soften the paper's tone instead of implicating anyone who reported an error with the infinite uncertainty specification of a Normal distribution and the authors could only bring themselves to go as far as a confidence that they are right in most cases. I cannot say I share their confidence.

The investigation of assumed priors given assumed likelihoods and then looking at the impact of different priors is not uninformative for the community. The authors could go further and use different likelihood interpretations, as these are just as arbitrary and subjective. If the message is " if you are Bayesian then think about your priors" then great. But I guess the message is more "everyone is acting like Bayesians with this model and these priors and that seems silly to us. Now everyone should be Bayesian and then justify their priors" (page 16 line 12). I am as Bayesian as they come and I think if researchers can be convinced that it is the only philosophically sound intellectual position, then they should be Bayesian and then they should think about their priors instead of using default priors. However, you have to first understand and accept the

subjectivist view of uncertainty and to state that the measurement model represents a full joint probability distribution over the data given the parameters (for you). Many and arguably most would choose not to do this. I've mentioned alternative views of the measurement model above and there are many other intellectually defensible and more mainstream views of the meaning of uncertainty statements. The paper and conclusions could easily have been softened to say "under the subjective Bayesian view, this would mean this and that would mean that" without then adding "it is our experience that almost everyone does mean this", and without even offering any of the alternative positions.

If we are being truly subjective, it seems hard to imagine that any researcher could truly hold a Cauchy prior. The huge tails actually make them less natural for things we can think about, such as a temperature change, as they assign much more probability to huge physically impossible changes as a distribution like the normal. It would not be hard for most Cauchy priors to find implied subjective probability statements that no researcher could agree with just from the cdf of the prior predictive distribution. This would not be a problem for an objectivist, but of course it is against the message of this paper. Given the strong message the authors wish to impart regarding careful consideration of the implications of their prior, the prior predictive should have been explored in more detail.

In summary, the authors have barely moved an inch from the first edition and my view of the paper has done similar.

[revised manuscript text omitted]